# Learning Invariant Representations for Reinforcement Learning without Reconstruction

**Amy Zhang**[*12]     **Rowan McAllister**[*3]     **Roberto Calandra**[2]     **Yarin Gal**[4]     **Sergey Levine**[3]
[1]McGill University
[2]Facebook AI Research
[3]University of California, Berkeley
[4]OATML group, University of Oxford

## Abstract

We study how representation learning can accelerate reinforcement learning from rich observations, such as images, without relying either on domain knowledge or pixel-reconstruction. Our goal is to learn representations that provide for effective downstream control and invariance to task-irrelevant details. Bisimulation metrics quantify behavioral similarity between states in continuous MDPs, which we propose using to learn robust latent representations which encode only the task-relevant information from observations. Our method trains encoders such that distances in latent space equal bisimulation distances in state space. We demonstrate the effectiveness of our method at disregarding task-irrelevant information using modified visual MuJoCo tasks, where the background is replaced with moving distractors and natural videos, while achieving SOTA performance. We also test a first-person highway driving task where our method learns invariance to clouds, weather, and time of day. Finally, we provide generalization results drawn from properties of bisimulation metrics, and links to causal inference.

## 1 Introduction

Learning control from images is important for many real world applications. While deep reinforcement learning (RL) has enjoyed many successes in simulated tasks, learning control from real vision is more complex, especially outdoors, where images reveal detailed scenes of a complex and unstructured world. Furthermore, while many RL algorithms can *eventually* learn control from real images given unlimited data, data-efficiency is often a necessity in real trials which are expensive and constrained to real-time. Prior methods for data-efficient learning of simulated visual tasks typically use representation learning. Representation learning summarizes images by encoding them into smaller vectored representations better suited for RL. For example, sequential autoencoders aim to learn *lossless* representations of streaming observations—sufficient to reconstruct current observations and predict future observations—from which various RL algorithms can be trained (Hafner et al., 2018; Lee et al., 2019; Yarats et al., 2019). However, such methods are *task-agnostic*: the models represent all dynamic elements they observe in the world, whether they are relevant to the task or not. We argue such representations can easily "distract" RL algorithms with irrelevant information in the case of real images. The issues of distraction is

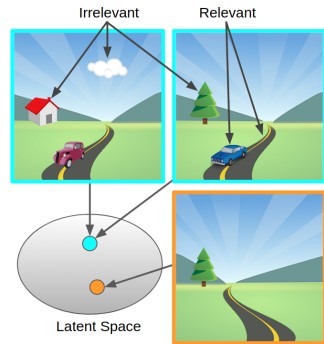

Figure 1: Robust representations of the visual scene should be insensitive to irrelevant objects (e.g., clouds) or details (e.g., car types), and encode two observations equivalently if their relevant details are equal (e.g., road direction and locations of other cars).

less evident in popular simulation MuJoCo and Atari tasks, since any change in observation space is likely task-relevant, and thus, worth representing. By contrast, visual images that autonomous cars observe contain predominantly task-irrelevant information, like cloud shapes and architectural details, illustrated in Figure 1.

---

[*]Equal contribution. Corresponding author: amy.x.zhang@mail.mcgill.ca

Rather than learning control-agnostic representations that focus on accurate reconstruction of clouds and buildings, we would rather achieve a more compressed representation from a *lossy* encoder, which only retains state information relevant to our task. If we would like to learn representations that capture only task-relevant elements of the state and are *invariant* to task-irrelevant information, intuitively we can utilize the reward signal to help determine task-relevance, as shown by Jonschkowski & Brock (2015). As *cumulative* rewards are our objective, state elements are relevant not only if they influence the current reward, but also if they influence state elements in the future that *in turn* influence future rewards. This recursive relationship can be distilled into a recursive task-aware notion of state abstraction: an ideal representation is one that is predictive of reward, and also predictive of itself in the future.

We propose learning such an invariant representation using the bisimulation metric, where the distance between two observation encodings correspond to how "behaviourally different" (Ferns & Precup, 2014) both observations are. Our main contribution is a practical representation learning method based on the bisimulation metric suitable for downstream control, which we call deep bisimulation for control (DBC). We additionally provide theoretical analysis that proves value bounds between the optimal value function of the true MDP and the optimal value function of the MDP constructed by the learned representation. Empirical evaluations demonstrate our non-reconstructive approach using bisimulation is substantially more robust to task-irrelevant distractors when compared to prior approaches that use reconstruction losses or contrastive losses. Our initial experiments insert natural videos into the background of MoJoCo control task as complex distraction. Our second setup is a high-fidelity highway driving task using CARLA (Dosovitskiy et al., 2017), showing that our representations can be trained effectively even on highly realistic images with many distractions, such as trees, clouds, buildings, and shadows. For example videos see `https://sites.google.com/view/deepbisim4control`.

## 2    Related Work

Our work builds on the extensive prior research on bisimulation in MDP state aggregation.

**Reconstruction-based Representations.** Early works on deep reinforcement learning from images (Lange & Riedmiller, 2010; Lange et al., 2012) used a two-step learning process where first an auto-encoder was trained using reconstruction loss to learn a low-dimensional representation, and subsequently a controller was learned using this representation. This allows effective leveraging of large, unlabeled datasets for learning representations for control. In practice, there is no guarantee that the learned representation will capture useful information for the control task, and significant expert knowledge and tricks are often necessary for these approaches to work. In model-based RL, one solution to this problem has been to jointly train the encoder and the dynamics model end-to-end (Watter et al., 2015; Wahlström et al., 2015) – this proved effective in learning useful task-oriented representations. Hafner et al. (2018) and Lee et al. (2019) learn latent state models using a reconstruction loss, but these approaches suffer from the difficulty of learning accurate long-term predictions and often still require significant manual tuning. Gelada et al. (2019) also propose a latent dynamics model-based method and connect their approach to bisimulation metrics, using a reconstruction loss in Atari. They show that $\ell_2$ distance in the DeepMDP representation upper bounds the bisimulation distance, whereas our objective directly learns a representation where distance in latent space *is* the bisimulation metric. Further, their results rely on the assumption that the learned representation is Lipschitz, whereas we show that, by directly learning a bisimilarity-based representation, we guarantee a representation that generates a Lipschitz MDP. We show experimentally that our *non-reconstructive* DBC method is substantially more robust to complex distractors.

**Contrastive-based Representations.** Contrastive losses are a self-supervised approach to learn useful representations by enforcing similarity constraints between data (van den Oord et al., 2018; Chen et al., 2020). Similarity functions can be provided as domain knowledge in the form of heuristic data augmentation, where we maximize similarity between augmentations of the same data point (Laskin et al., 2020) or nearby image patches (Hénaff et al., 2019), and minimize similarity between different data points. In the absence of this domain knowledge, contrastive representations can be trained by predicting the future (van den Oord et al., 2018). We compare to such an approach in our experiments, and show that DBC is substantially more robust. While contrastive losses do not require reconstruction, they do not inherently have a mechanism to determine downstream task relevance without manual engineering, and when trained only for prediction, they aim to capture all

predictable features in the observation, which performs poorly on real images for the same reasons world models do. A better method would be to incorporate knowledge of the downstream task into the similarity function in a data-driven way, so that images that are very different pixel-wise (e.g. lighting or texture changes), can also be grouped as similar w.r.t. downstream objectives.

**Bisimulation.** Various forms of state abstractions have been defined in Markov decision processes (MDPs) to group states into clusters whilst preserving some property (e.g. the optimal value, or all values, or all action values from each state) (Li et al., 2006). The strictest form, which generally preserves the most properties, is *bisimulation* (Larsen & Skou, 1989). Bisimulation only groups states that are indistinguishable w.r.t. reward sequences output given any action sequence tested. A related concept is bisimulation metrics (Ferns & Precup, 2014), which measure how "behaviorally similar" states are. Ferns et al. (2011) defines the bisimulation metric with respect to continuous MDPs, and propose a Monte Carlo algorithm for learning it using an exact computation of the Wasserstein distance between empirically measured transition distributions. However, this method does not scale well to large state spaces. Taylor et al. (2009) relate MDP homomorphisms to lax probabilistic bisimulation, and define a lax bisimulation metric. They then compute a value bound based on this metric for MDP homomorphisms, where approximately equivalent state-action pairs are aggregated. Most recently, Castro (2020) propose an algorithm for computing *on-policy* bisimulation metrics, but does so directly, without learning a representation. They focus on deterministic settings and the policy evaluation problem. We believe our work is the first to propose a gradient-based method for directly learning a *representation space* with the properties of bisimulation metrics and show that it works in the policy optimization setting.

## 3 Preliminaries

We start by introducing notation and outlining realistic assumptions about underlying structure in the environment. Then, we review state abstractions and metrics for state similarity.

We assume the underlying environment is a **Markov decision process** (MDP), described by the tuple $\mathcal{M} = (\mathcal{S}, \mathcal{A}, \mathcal{P}, \mathcal{R}, \gamma)$, where $\mathcal{S}$ is the state space, $\mathcal{A}$ the action space, $\mathcal{P}(\mathbf{s}'|\mathbf{s}, \mathbf{a})$ the probability of transitioning from state $\mathbf{s} \in \mathcal{S}$ to state $\mathbf{s}' \in \mathcal{S}$, and $\gamma \in [0, 1)$ a discount factor. An "agent" chooses actions $\mathbf{a} \in \mathcal{A}$ according to a policy function $\mathbf{a} \sim \pi(\mathbf{s})$, which updates the system state $\mathbf{s}' \sim \mathcal{P}(\mathbf{s}, \mathbf{a})$, yielding a reward $r = \mathcal{R}(\mathbf{s}) \in \mathbb{R}$. The agent's goal is to maximize the expected cumulative discounted rewards by learning a good policy: $\max_\pi \mathbb{E}_\mathcal{P}[\sum_{t=0}^{\infty}[\gamma^t \mathcal{R}(\mathbf{s}_t)]]$. While our primary concern is learning from images, we do not address the partial-observability problem explicitly: we instead approximate *stacked* pixel observations as the fully-observed system state $\mathbf{s}$ (explained further in Appendix B).

**Bisimulation** is a form of state abstraction that groups states $\mathbf{s}_i$ and $\mathbf{s}_j$ that are "behaviorally equivalent" (Li et al., 2006). For any action sequence $\mathbf{a}_{0:\infty}$, the probabilistic sequence of rewards from $\mathbf{s}_i$ and $\mathbf{s}_j$ are identical. A more compact definition has a recursive form: two states are bisimilar if they share both the same immediate reward and equivalent distributions over the next bisimilar states (Larsen & Skou, 1989; Givan et al., 2003).

**Definition 1** (Bisimulation Relations (Givan et al., 2003)). *Given an MDP $\mathcal{M}$, an equivalence relation $B$ between states is a bisimulation relation if, for all states $\mathbf{s}_i, \mathbf{s}_j \in \mathcal{S}$ that are equivalent under $B$ (denoted $\mathbf{s}_i \equiv_B \mathbf{s}_j$) the following conditions hold:*

$$\mathcal{R}(\mathbf{s}_i, \mathbf{a}) = \mathcal{R}(\mathbf{s}_j, \mathbf{a}) \qquad \forall \mathbf{a} \in \mathcal{A}, \tag{1}$$

$$\mathcal{P}(G|\mathbf{s}_i, \mathbf{a}) = \mathcal{P}(G|\mathbf{s}_j, \mathbf{a}) \quad \forall \mathbf{a} \in \mathcal{A}, \quad \forall G \in \mathcal{S}_B, \tag{2}$$

*where $\mathcal{S}_B$ is the partition of $\mathcal{S}$ under the relation $B$ (the set of all groups $G$ of equivalent states), and $\mathcal{P}(G|\mathbf{s}, \mathbf{a}) = \sum_{\mathbf{s}' \in G} \mathcal{P}(\mathbf{s}'|\mathbf{s}, \mathbf{a})$.*

Exact partitioning with bisimulation relations is generally impractical in continuous state spaces, as the relation is highly sensitive to infinitesimal changes in the reward function or dynamics. For this reason, **Bisimulation Metrics** (Ferns et al., 2011; Ferns & Precup, 2014; Castro, 2020) softens the concept of state partitions, and instead defines a pseudometric space $(\mathcal{S}, d)$, where a distance function $d : \mathcal{S} \times \mathcal{S} \mapsto \mathbb{R}_{\geq 0}$ measures the "behavioral similarity" between two states[1].

Defining a distance $d$ between states requires defining both a distance between rewards (to soften Equation (1)), and distance between state distributions (to soften Equation (2)). Prior works use the Wasserstein metric for the latter, originally used in the context of bisimulation metrics by van Breugel

---

[1]Note that $d$ is a pseudometric, meaning the distance between two different states can be zero, corresponding to behavioral equivalence.

& Worrell (2001). The $p^{\text{th}}$ Wasserstein metric is defined between two probability distributions $\mathcal{P}_i$ and $\mathcal{P}_j$ as $W_p(\mathcal{P}_i, \mathcal{P}_j; d) = (\inf_{\gamma' \in \Gamma(\mathcal{P}_i, \mathcal{P}_j)} \int_{\mathcal{S} \times \mathcal{S}} d(\mathbf{s}_i, \mathbf{s}_j)^p \, \mathrm{d}\gamma'(\mathbf{s}_i, \mathbf{s}_j))^{1/p}$, where $\Gamma(\mathcal{P}_i, \mathcal{P}_j)$ is the set of all couplings of $\mathcal{P}_i$ and $\mathcal{P}_j$. This is known as the "earth mover" distance, denoting the cost of transporting mass from one distribution to another (Villani, 2003). Finally, the bisimulation metric is the reward difference added to the Wasserstein distance between transition distributions:

**Definition 2** (Bisimulation Metric). *From Theorem 2.6 in Ferns et al. (2011) with $c \in [0, 1)$:*
$$d(\mathbf{s}_i, \mathbf{s}_j) \;=\; \max_{\mathbf{a} \in \mathcal{A}} (1 - c) \cdot |\mathcal{R}_{\mathbf{s}_i}^{\mathbf{a}} - \mathcal{R}_{\mathbf{s}_j}^{\mathbf{a}}| + c \cdot W_1(\mathcal{P}_{\mathbf{s}_i}^{\mathbf{a}}, \mathcal{P}_{\mathbf{s}_j}^{\mathbf{a}}; d). \tag{3}$$

# 4 Learning Representations for Control with Bisimulation Metrics

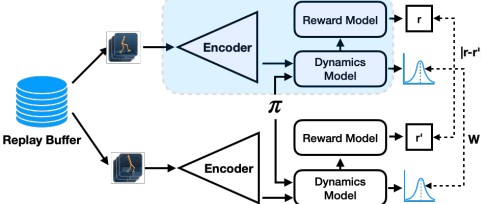

Figure 2: Learning a bisimulation metric representation: shaded in blue is the main model architecture, it is reused for both states, like a Siamese network. The loss is the reward and discounted transition distribution distances (using Wasserstein metric $W$).

**Algorithm 1** Deep Bisimulation for Control (DBC)

1: **for** Time $t = 0$ to $\infty$ **do**
2:     Encode state $\mathbf{z}_t = \phi(\mathbf{s}_t)$
3:     Execute action $\mathbf{a}_t \sim \pi(\mathbf{z}_t)$
4:     Record data: $\mathcal{D} \leftarrow \mathcal{D} \cup \{\mathbf{s}_t, \mathbf{a}_t, \mathbf{s}_{t+1}, r_{t+1}\}$
5:     Sample batch $B_i \sim \mathcal{D}$
6:     Permute batch: $B_j = \text{permute}(B_i)$
7:     Train policy: $\mathbb{E}_{B_i}[J(\pi)]$     ▷ Algorithm 2
8:     Train encoder: $\mathbb{E}_{B_i, B_j}[J(\phi)]$     ▷ Equation (4)
9:     Train dynamics: $J(\hat{\mathcal{P}}, \phi) = (\hat{\mathcal{P}}(\phi(\mathbf{s}_t), \mathbf{a}_t) - \bar{\mathbf{z}}_{t+1})^2$

We propose Deep Bisimulation for Control (DBC), a data-efficient approach to learn control policies from unstructured, high-dimensional states. In contrast to prior work on bisimulation, which typically aims to learn a distance function of the form $d : \mathcal{S} \times \mathcal{S} \mapsto \mathbb{R}_{\geq 0}$ between states, our aim is instead to learn *representations* $\mathcal{Z}$ under which $\ell_1$ distances correspond to bisimulation metrics, and then use these representations to improve reinforcement learning. Our goal is to learn encoders $\phi : \mathcal{S} \mapsto \mathcal{Z}$ that capture representations of states that are suitable to control, while discarding any information that is *irrelevant* for control. Any representation that relies on reconstruction of the state cannot do this, as these irrelevant details are still important for reconstruction. We hypothesize that bisimulation metrics can acquire this type of representation, without any reconstruction.

Bisimulation metrics are a useful form of state abstraction, but prior methods to train distance functions either do not scale to pixel observations (Ferns et al., 2011) (due to the max operator in Equation (3)), or were only designed for the (fixed) policy evaluation setting (Castro, 2020). By contrast, we learn improved representations for policy inputs, as the policy improves online. Our $\pi^*$-*bisimulation metric* is learned with gradient decent, and we prove it converges to a fixed point in Theorem 1 under some assumptions. To train our encoder $\phi$ towards our desired relation $d(\mathbf{s}_i, \mathbf{s}_j) := ||\phi(\mathbf{s}_i) - \phi(\mathbf{s}_j)||_1$, we draw batches of state pairs, and minimise the mean square error between the on-policy bisimulation metric and $\ell_1$ distance in the latent space:

$$J(\phi) \;=\; \left( ||\mathbf{z}_i - \mathbf{z}_j||_1 \;-\; |r_i - r_j| \;-\; \gamma W_2\big(\hat{\mathcal{P}}(\cdot|\bar{\mathbf{z}}_i, \mathbf{a}_i), \hat{\mathcal{P}}(\cdot|\bar{\mathbf{z}}_j, \mathbf{a}_j)\big) \right)^2, \tag{4}$$

where $\mathbf{z}_i = \phi(\mathbf{s}_i)$, $\mathbf{z}_j = \phi(\mathbf{s}_j)$, $r$ are rewards, and $\bar{\mathbf{z}}$ denotes $\phi(\mathbf{s})$ with stop gradients. Equation (4) also uses a probabilistic dynamics model $\hat{\mathcal{P}}$ which outputs a Gaussian distribution. For this reason, we use the 2-Wasserstein metric $W_2$ in Equation (4), as opposed to the 1-Wasserstein in Equation (3), since the $W_2$ metric has a convenient closed form: $W_2(\mathcal{N}(\mu_i, \Sigma_i), \mathcal{N}(\mu_j, \Sigma_j))^2 = ||\mu_i - \mu_j||_2^2 + ||\Sigma_i^{1/2} - \Sigma_j^{1/2}||_{\mathcal{F}}^2$, where $|| \cdot ||_{\mathcal{F}}$ is the Frobenius norm. For all other distances we continue using the $\ell_1$ norm. Our model architecture and training is illustrated by Figure 2 and Algorithm 1.

**Incorporating control.** We combine our representation learning approach (Algorithm 1) with the soft actor-critic (SAC) algorithm (Haarnoja et al., 2018) to devise a practical reinforcement learning method. We modified SAC slightly in Algorithm 2 to allow the value function to backprop to our encoder, which can improve performance further (Yarats et al.,

**Algorithm 2** Train Policy (changes to SAC in blue)

1: Get value: $V = \min_{i=1,2} \hat{Q}_i(\hat{\phi}(\mathbf{s})) - \alpha \log \pi(\mathbf{a}|\phi(\mathbf{s}))$
2: Train critics: $J(Q_i, \phi) = (Q_i(\phi(\mathbf{s})) - r - \gamma V)^2$
3: Train actor: $J(\pi) = \alpha \log p(\mathbf{a}|\phi(\mathbf{s})) - \min_{i=1,2} Q_i(\phi(\mathbf{s}))$
4: Train alpha: $J(\alpha) = -\alpha \log p(\mathbf{a}|\phi(\mathbf{s}))$
5: Update target critics: $\hat{Q}_i \leftarrow \tau_Q Q_i + (1 - \tau_Q)\hat{Q}_i$
6: Update target encoder: $\hat{\phi} \leftarrow \tau_\phi \phi + (1 - \tau_\phi)\hat{\phi}$

2019; Rakelly et al., 2019). Although, in principle, our method could be combined with any RL algorithm, including the model-free DQN (Mnih et al., 2015), or model-based PETS (Chua et al.,

2018). Implementation details and hyperparameter values of DBC are summarized in the appendix, Table 2. We train DBC by iteratively updating three components in turn: a policy $\pi$ (in this case SAC), an encoder $\phi$, and a dynamics model $\mathcal{P}$ (lines 7–9, Algorithm 1). We found a single loss function was less stable to train. The inputs of each loss function $J(\cdot)$ in Algorithm 1 represents which components are updated. After each training step, the policy $\pi$ is used to step in the environment, the data is collected in a replay buffer $\mathcal{D}$, and a batch is randomly selected to repeat training.

## 5  Generalization Bounds and Links to Causal Inference

While DBC enables representation learning without pixel reconstruction, it leaves open the question of how good the resulting representations really are. In this section, we present theoretical analysis that bounds the suboptimality of a value function trained on the representation learned via DBC. First, we show that our $\pi^*$-*bisimulation metric* converges to a fixed point, starting from the initialized policy $\pi_0$ and converging to an optimal policy $\pi^*$.

**Theorem 1.** *Let* $\mathfrak{met}$ *be the space of bounded pseudometrics on* $\mathcal{S}$ *and* $\pi$ *a policy that is continuously improving. Define* $\mathcal{F} : \mathfrak{met} \mapsto \mathfrak{met}$ *by*

$$\mathcal{F}(d, \pi)(\mathbf{s}_i, \mathbf{s}_j) = (1 - c)|r_{\mathbf{s}_i}^\pi - r_{\mathbf{s}_j}^\pi| + cW(d)(\mathcal{P}_{\mathbf{s}_i}^\pi, \mathcal{P}_{\mathbf{s}_j}^\pi). \tag{5}$$

*Then* $\mathcal{F}$ *has a least fixed point* $\tilde{d}$ *which is a* $\pi^*$-*bisimulation metric.*

Proof in appendix. As evidenced by Definition 2, the bisimulation metric has no direct dependence on the state space. Pixels can change, but bisimilarity will stay the same. Instead, bisimilarity is grounded in a recursion of future transition probabilities and rewards, which is closely related to the optimal value function. In fact, the bisimulation metric gives tight bounds on the optimal value function with discount factor $\gamma$. We show this using the property that the optimal value function is Lipschitz with respect to the bisimulation metric, see Theorem 5 in Appendix (Ferns et al., 2004). This result also implies that the closer two states are in terms of $\tilde{d}$, the more likely they are to share the same optimal actions. This leads us to a generalization bound on the optimal value function of an MDP constructed from a representation space using bisimulation metrics, $||\phi(\mathbf{s}_i) - \phi(\mathbf{s}_j)||_1 := \tilde{d}(\mathbf{s}_i, \mathbf{s}_j)$. We can construct a partition of this space for some $\epsilon > 0$, giving us $n$ partitions where $\frac{1}{n} < (1 - c)\epsilon$. We denote $\phi$ as the encoder that maps from the original state space $\mathcal{S}$ to each $\epsilon$-cluster. This $\epsilon$ denotes the amount of approximation allowed, where large $\epsilon$ leads to a more compact bisimulation partition at the expense of a looser bound on the optimal value function.

**Theorem 2** (Value bound based on bisimulation metrics). *Given an MDP* $\bar{\mathcal{M}}$ *constructed by aggregating states in an* $\epsilon$-*neighborhood, and an encoder* $\phi$ *that maps from states in the original MDP* $\mathcal{M}$ *to these clusters, the optimal value functions for the two MDPs are bounded as*

$$|V^*(\mathbf{s}) - V^*(\phi(\mathbf{s}))| \leq \frac{2\epsilon}{(1 - \gamma)(1 - c)}. \tag{6}$$

Proof in appendix. As $\epsilon \to 0$ the optimal value function of the aggregated MDP converges to the original. Further, by defining a learning error for $\phi$, $\mathcal{L} := \sup_{\mathbf{s}_i, \mathbf{s}_j \in \mathcal{S}} \left| ||\phi(\mathbf{s}_i) - \phi(\mathbf{s}_j)||_1 - \tilde{d}(\mathbf{s}_i, \mathbf{s}_j) \right|$, we can update the bound in Theorem 2 to incorporate $\mathcal{L}$: $|V^*(\mathbf{s}) - V^*(\phi(\mathbf{s}))| \leq \frac{2\epsilon + 2\mathcal{L}}{(1 - \gamma)(1 - c)}$.

MDP dynamics have a strong connection to causal inference and causal graphs, which are directed acyclic graphs (Jonsson & Barto, 2006; Schölkopf, 2019; Zhang et al., 2020). Specifically, the state and action at time $t$ causally affect the next state at time $t + 1$. In this work, we care about the components of the state space that causally affect current and future reward. Deep bisimulation for control representations connect to *causal feature sets*, or the minimal feature set needed to predict a target variable (Zhang et al., 2020).

**Theorem 3** (Connections to causal feature sets (Thm 1 in Zhang et al. (2020))). *If we partition observations using the bisimulation metric, those clusters (a bisimulation partition) correspond to the causal feature set of the observation space with respect to current and future reward.*

This connection tells us that these features are the minimal sufficient statistic of the current and future reward, and therefore consist of (and only consist of) the *causal ancestors* of the reward variable $r$.

**Definition 3** (Causal Ancestors). *In a causal graph where nodes correspond to variables and directed edges between a parent node* $P$ *and child node* $C$ *are causal relationships, the causal ancestors* $AN(C)$ *of a node are all nodes in the path from* $C$ *to a root node.*

If there are interventions on *distractor variables*, or variables that control the rendering function $q$ and therefore the rendered observation but do not affect the reward, the causal feature set will be

robust to these interventions, and correctly predict current and future reward in the linear function approximation setting (Zhang et al., 2020). As an example, in autonomous driving, an intervention can be a change from day to night which affects the observation space but not the dynamics or reward. Finally, we show that a representation based on the bisimulation metric generalizes to other reward functions with the same causal ancestors.

**Theorem 4** (Task Generalization). *Given an encoder* $\phi : \mathcal{S} \mapsto \mathcal{Z}$ *that maps observations to a latent bisimulation metric representation where* $||\phi(\mathbf{s}_i) - \phi(\mathbf{s}_j)||_1 := \tilde{d}(\mathbf{s}_i, \mathbf{s}_j)$, $\mathcal{Z}$ *encodes information about all the causal ancestors of the reward* $AN(R)$.

Proof in appendix. This result shows that the learned representation will generalize to unseen reward functions, as long as the new reward function has a subset of the same causal ancestors. As an example, a representation learned for a robot to walk will likely generalize to learning to run, because the reward function depends on forward velocity and all the factors that contribute to forward velocity. However, that representation will not generalize to picking up objects, as those objects will be ignored by the learned representation, since they are not likely to be causal ancestors of a reward function designed for walking. Theorem 4 shows that the learned representation will be robust to spurious correlations, or changes in factors that are not in $AN(R)$. This complements Theorem 5, that the representation is a minimal sufficient statistic of the optimal value function, improving generalization over non-minimal representations.

**Theorem 5** ($V^*$ is Lipschitz with respect to $\tilde{d}$). *Let* $V^*$ *be the optimal value function for a given discount factor* $\gamma$. *If* $c \geq \gamma$, *then* $V^*$ *is Lipschitz continuous with respect to* $\tilde{d}$ *with Lipschitz constant* $\frac{1}{1-c}$, *where* $\tilde{d}$ *is a* $\pi^*$*-bisimulation metric.*

$$|V^*(\mathbf{s}_i) - V^*(\mathbf{s}_j)| \leq \frac{1}{1-c}\tilde{d}(\mathbf{s}_i, \mathbf{s}_j). \tag{7}$$

See Theorem 5.1 in Ferns et al. (2004) for proof. We show empirical validation of these findings in Section 6.2.

# 6 Experiments

Our central hypothesis is that our non-reconstructive bisimulation based representation learning approach should be substantially more robust to task-irrelevant distractors. To that end, we evaluate our method in a clean setting without distractors, as well as a much more difficult setting with distractors. We compare against several baselines. The first is Stochastic Latent Actor-Critic (SLAC, Lee et al. (2019)), a state-of-the-art method for pixel observations on DeepMind Control that learns a dynamics model with a reconstruction loss. The second is DeepMDP (Gelada et al., 2019), a recent method that also learns a latent representation space using a latent dynamics model, reward model, and distributional Q learning, but for which they needed a reconstruction loss to scale up to Atari. Finally, we compare against two methods using the same architecture as ours but exchange our bisimulation loss with (1) a reconstruction loss ("*Reconstruction*") and (2) contrastive predictive coding (Oord et al., 2018) ("*Contrastive*") to ground the dynamics model and learn a latent representation.

## 6.1 Control with Background Distraction

In this section, we benchmark DBC and the previously described baselines on the DeepMind Control (DMC) suite (Tassa et al., 2018) in two settings and nine environments (Figure 3), `finger_spin`, `cheetah_run`, and `walker_walk` and additional environments in the appendix.

**Default Setting.** Here, the pixel observations have simple backgrounds as shown in Figure 3 (top row) with training curves for our DBC and baselines. We see SLAC, a recent state-of-the-art model-based representation learning method that uses reconstruction, generally performs best.

**Simple Distractors Setting.** Next, we include simple background distractors, shown in Figure 3 (middle row), with easy-to-predict motions. We use a fixed number of colored circles that obey the dynamics of an ideal gas (no attraction or repulsion between objects) with no collisions. Note the performance of DBC remains consistent, as other methods start decreasing.

**Natural Video Setting.** Then, we incorporate natural video from the Kinetics dataset (Kay et al., 2017) as background (Zhang et al., 2018), shown in Figure 3 (bottom row). The results confirm our hypothesis: although a number of prior methods can learn effectively in the absence of distractors, when complex distractions are introduced, our non-reconstructive bisimulation based method attains substantially better results.

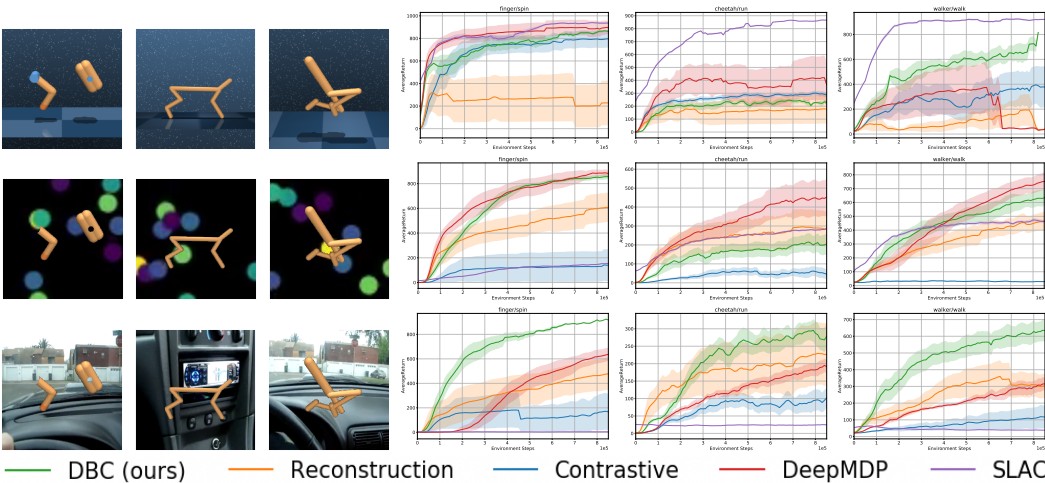

Figure 3: **Left observations**: Pixel observations in DMC in the default setting (top row) of the finger spin (left column), cheetah (middle column), and walker (right column), with simple distractors (middle row), and natural video distractors (bottom row). **Right training curves**: Results comparing out DBC method to baselines on 10 seeds with 1 standard error shaded in the default setting. The grid-location of each graph corresponds to the grid-location of each observation.

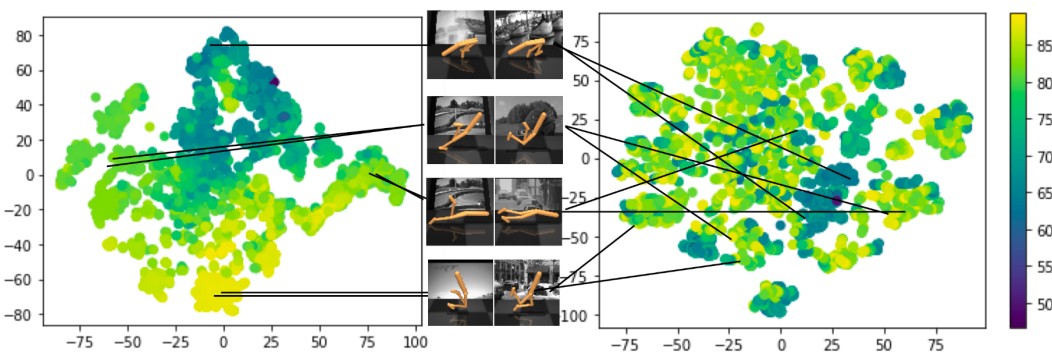

Figure 4: t-SNE of latent spaces learned with a bisimulation metric (left t-SNE) and VAE (right t-SNE) after training has completed, color-coded with predicted state values (higher value yellow, lower value purple). Neighboring points in the embedding space learned with a bisimulation metric have similar states and correspond to observations with the same task-related information (depicted as pairs of images with their corresponding embeddings), whereas no such structure is seen in the embedding space learned by VAE, where the same image pairs are mapped far away from each other.

To visualize the representation learned with our bisimulation metric loss function in Equation (4), we use a t-SNE plot (Figure 4). We see that even when the background looks drastically different, our encoder learns to ignore irrelevant information and maps observations with similar robot configurations near each other. See Appendix D for another visualization.

## 6.2 Generalization Experiments

We test generalization of our learned representation in two ways. First, we show that the learned representation space can generalize to different types of distractors, by training with simple distractors and testing on the natural video setting. Second, we show that our learned representation can be useful reward functions other than those it was trained for.

**Generalizing over backgrounds.** We first train on the `simple distractors` setting and evaluate on `natural video`. Figure 5 shows an example of the `simple distractors` setting and performance during training time of two experiments, blue being the zero-shot transfer to the `natural video` setting, and orange the baseline which trains on `natural video`. This result empirically validates that the representations learned by DBC are able to effectively learn to ignore the background, *regardless* of what the background contains or how dynamic it is.

**Generalizing over reward functions.** We evaluate (Figure 5) the generalization capabilities of the learned representation by training SAC with new reward functions `walker_stand` and `walker_run` using the fixed representation learned from `walker_walk`. This is empirical evidence that confirms Theorem 4: if the new reward functions are causally dependent on a subset of the same factors that determine the original reward function, then our representation is sufficient.

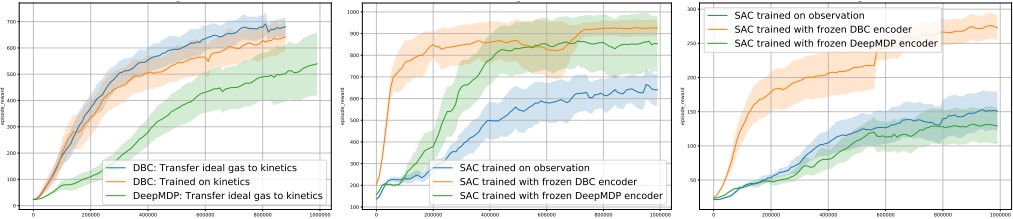

Figure 5: Generalization of a model trained on `simple distractors` environment and evaluated on `kinetics` (left). Generalization of an encoder trained on `walker_walk` environment and evaluated on `walker_stand` (center) and `walker_run` (right), all in the `simple distractors` setting. 10 seeds, 1 standard error shaded.

## 6.3 Comparison with other Bisimulation Encoders

Even though the purpose of bisimulation metrics by Castro (2020) is learning distances $d$, not representation spaces $\mathcal{Z}$, it nevertheless implements $d$ with function approximation: $d(\mathbf{s}_i, \mathbf{s}_j) = \psi\big(\phi(\mathbf{s}_i), \phi(\mathbf{s}_j)\big)$ by encoding observations with $\phi$ before computing distances with $\psi$, trained as:

$$J(\phi, \psi) = \Big( \psi\big(\phi(\mathbf{s}_i), \phi(\mathbf{s}_j)\big) - |r_i - r_j| - \gamma\hat{\psi}\big(\hat{\phi}\big(\mathcal{P}(\mathbf{s}_i, \pi(\mathbf{s}_i))\big), \hat{\phi}\big(\mathcal{P}(\mathbf{s}_j, \pi(\mathbf{s}_j))\big)\big) \Big)^2, \quad (8)$$

where $\hat{\phi}$ and $\hat{\psi}$ are target networks. A natural question is: how does the encoder $\phi$ above perform in control tasks? We combine $\phi$ above with our policy in Algorithm 2 and use the same network $\psi$ (single hidden layer 729 wide). Figure 6 shows representations from Castro (2020) can learn control (surprisingly well given it was not designed to), but our method learns faster. Further, our method is simpler: by comparing Equation (8) to Equation (4), our method uses the $\ell_1$ distance between the encoding instead of introducing an addition network $\psi$.

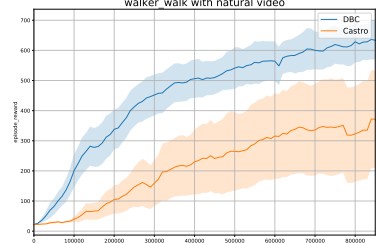

Figure 6: Bisim. results. Blue is DBC and orange is Castro (2020).

## 6.4 Autonomous Driving with Visual Redundancy

Real-world control systems such as robotics and autonomous vehicles must contend with a huge variety of task-irrelevant information, such as irrelevant *objects* (e.g. clouds) and irrelevant *details* (e.g. obstacle color). To evaluate DBC on tasks with more realistic observations, we construct a highway driving scenario with photo-realistic visual observations

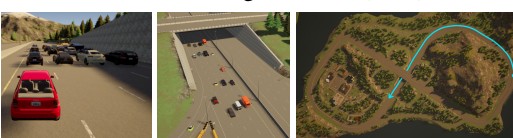

Figure 7: The driving task is to drive the red ego car (left) safely in traffic (middle) along a highway (right).

using the CARLA simulator (Dosovitskiy et al., 2017) shown in Figure 7. The agent's goal is to drive as far as possible along CARLA's Town04's figure-8 the highway in 1000 time-steps without colliding into the 20 other moving vehicles or barriers. Our objective function rewards highway progression and penalises collisions: $r_t = \mathbf{v}_{\text{ego}}^{\top}\hat{\mathbf{u}}_{\text{highway}} \cdot \Delta t - \lambda_i \cdot \text{impulse} - \lambda_s \cdot |\text{steer}|$, where $\mathbf{v}_{\text{ego}}$ is the velocity vector of the ego vehicle, projected onto the highway's unit vector $\hat{\mathbf{u}}_{\text{highway}}$, and multiplied by time discretization $\Delta t = 0.05$ to measure highway progression in meters. Collisions result in impulses $\in \mathbb{R}^+$, measured in Newton-seconds. We found a steering penalty steer $\in [-1, 1]$ helped, and used weights $\lambda_i = 10^{-4}$ and $\lambda_s = 1$. While more specialized objectives exist like lane-keeping, this experiment's purpose is only to compare representations with observations more characteristic of real robotic tasks. We use five cameras on the vehicle's roof, each with 60 degree views. By concatenating the images together, our vehicle has a 300 degree view, observed as $84 \times 420$ pixels. Code and install instructions in appendix.

**Results** in Figure 9 compare the same baselines as before, except for SLAC which is easily distracted (Figure 3). Instead we used SAC, which does not explicitly learn a representation, but performs surprisingly well from raw images. DeepMDP performs well too, perhaps given its similarly to bisimulation. But, Reconstruction and Contrastive methods again perform poorly with complex

images. More intuitive metrics are in Table 1 and Figure 8 depicts the representation space as a t-SNE with corresponding observations. Each run took 12 hours on a GTX 1080 GPU.

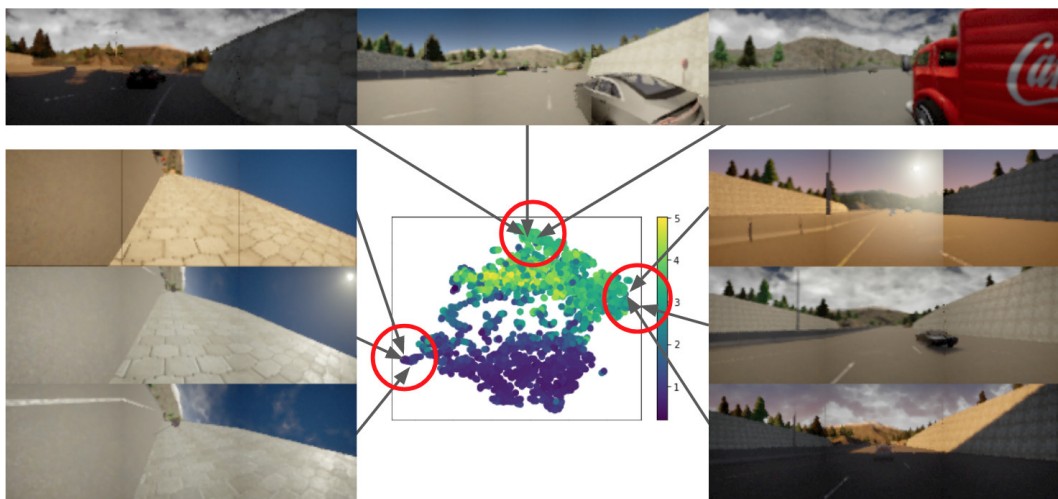

Figure 8: A t-SNE diagram of encoded first-person driving observations after 10k training steps of Algorithm 1, color coded by value ($V$ in Algorithm 2). **Top**: the learned representation identifies an obstacle on the right side. Whether that obstacle is a dark wall, bright car, or truck is task-irrelevant: these states are behaviourally equivalent. **Left**: the ego vehicle has flipped onto its left side. The different wall colors, due to a setting sun, is irrelevant: all states are equally stuck and low-value (purple t-SNE color). **Right**: clear highway driving. Clouds and sun position are irrelevant.

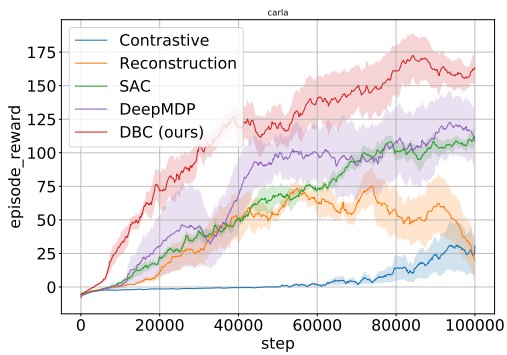

Figure 9: Performance comparison with 3 seeds on the driving task. Our DBC method (red) performs better than DeepMDP (purple) or learning direct from pixels without a representation (SAC, green), and much better than contrastive methods (blue). Our method's final performance is 46.8% better than the next best baseline.

Table 1: Driving metrics, averaged over 100 episodes, after 100k training steps, with standard error. Arrow direction indicates if metric desired larger or smaller.

| | | SAC | DeepMDP | **DBC (ours)** |
|---|---|---|---|---|
| successes (100m) | ↑ | 12% | 17% | **24%** |
| distance (m) | ↑ | $123.2 \pm 7.43$ | $106.7 \pm 11.1$ | **179.0** $\pm 11.4$ |
| crash intensity | ↓ | $4604 \pm 30.7$ | **1958** $\pm 15.6$ | $2673 \pm 38.5$ |
| average steer | ↓ | $16.6\% \pm 0.019\%$ | $10.4\% \pm 0.015\%$ | **7.3%** $\pm 0.012\%$ |
| average brake | ↓ | **1.3%** $\pm 0.006\%$ | $4.3\% \pm 0.033\%$ | $1.6\% \pm 0.022\%$ |

# 7 Discussion

This paper presents Deep Bisimulation for Control: a new representation learning method that considers downstream control. Observations are encoded into representations that are *invariant* to different task-irrelevant details in the observation. We show this is important when learning control from outdoor images, or otherwise images with background "distractions". In contrast to other bisimulation methods, we show performance gains when distances in representation space match the bisimulation distance between observations.

**Future work:** Several options exist for future work. First, our latent dynamics model $\hat{\mathcal{P}}$ was only used for training our encoder in Equation (4), but could also be used for multi-step planning in latent space. Second, estimating uncertainty could also be important to produce agents that can work in the real world, perhaps via an ensemble of models $\{\hat{\mathcal{P}}_k\}_{k=1}^{K}$, to detect—and adapt to—distributional shifts between training and test observations. Third, an undressed issue is that of partially observed settings (that assumed approximately full observability by using stacked images), possibly using explicit memory or implicit memory such as an LSTM. Finally, investigating which metrics (L1 or L2) and dynamics distributions (Gaussians or not) would be beneficial.

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

## A Additional Theorems and Proofs

**Theorem 1.** *Let* $\mathfrak{met}$ *be the space of bounded pseudometrics on $S$ and $\pi \in \Pi$ a policy that is continuously improving in the space of policies $\Pi$. Define $\mathcal{F} : \mathfrak{met} \times \Pi \mapsto \mathfrak{met}$ by*
$$\mathcal{F}(d, \pi)(\mathbf{s}_i, \mathbf{s}_j) = (1 - c)|r_{\mathbf{s}_i}^{\pi} - r_{\mathbf{s}_j}^{\pi}| + cW(d)(\mathcal{P}_{\mathbf{s}_i}^{\pi}, \mathcal{P}_{\mathbf{s}_j}^{\pi}). \tag{9}$$
*Then $\mathcal{F}$ has a least fixed point $\tilde{d}$ which is a $\pi^*$-bisimulation metric.*

*Proof.* Ideally, to prove this theorem we show that $\mathcal{F}$ is monotonically increasing and continuous, and apply Fixed Point Theorem to show the existence of a fixed point that $\mathcal{F}$ converges to. Unfortunately, we can show that $\mathcal{F}$ under $\pi$ as $\pi$ monotonically converges to $\pi^*$ is *not* also monotonic, unlike the original bisimulation metric setting (Ferns et al., 2004) and the policy evaluation setting (Castro, 2020). We start the iterates $\mathcal{F}^n$ from bottom $\bot$, denoted as $\mathcal{F}^n(\bot)$. In Ferns et al. (2004) the $\max_{\mathbf{a} \in \mathcal{A}}$ can be thought of as learning a policy between every two pairs of states to maximize their distance, and therefore this distance can only stay the same or grow over iterations of $\mathcal{F}$. In Castro (2020), $\pi$ is fixed, and under a deterministic MDP it can also be shown that distance between states $d_n(\mathbf{s}_i, \mathbf{s}_j)$ will only expand, not contract as $n$ increases. In the policy iteration setting, however, with $\pi$ starting from initialization $\pi_0$ and getting updated:
$$\pi_k(\mathbf{s}) = \arg\max_{\mathbf{a} \in \mathcal{A}} \sum_{\mathbf{s}' \in S} [r_{\mathbf{ss}'}^{\mathbf{a}} + \gamma V^{\pi_{k-1}}(\mathbf{s}')], \tag{10}$$
there is no guarantee that the distance between two states $d_{n-1}^{\pi_{k-1}}(\mathbf{s}_i, \mathbf{s}_j) < d_n^{\pi_k}(\mathbf{s}_i, \mathbf{s}_j)$ under policy iterations $\pi_{k-1}, \pi_k$ and distance metric iterations $d_{n-1}, d_n$ for $k, n \in \mathbb{N}$, which is required for monotonicity.

Instead, we show that using the policy improvement theorem which gives us
$$V^{\pi_k}(\mathbf{s}) \geq V^{\pi_{k-1}}(\mathbf{s}), \forall \mathbf{s} \in \mathcal{S}, \tag{11}$$
$\pi$ will converge to a fixed point using the Fixed Point Theorem, and taking the result by Castro (2020) that $\mathcal{F}^{\pi}$ has a fixed point for every $\pi \in \Pi$, we can show that a fixed point bisimulation metric will be found with policy iteration. $\square$

**Theorem 2.** *Given a new aggregated MDP $\bar{\mathcal{M}}$ constructed by aggregating states in an $\epsilon$-neighborhood, and an encoder $\phi$ that maps from states in the original MDP $\mathcal{M}$ to these clusters, the optimal value functions for the two MDPs are bounded as*
$$|V^*(\mathbf{s}) - V^*(\phi(\mathbf{s}))| \leq \frac{2\epsilon}{(1 - \gamma)(1 - c)}. \tag{12}$$

*Proof.* From Theorem 5.1 in Ferns et al. (2004) we have:
$$(1 - c)|V^*(\mathbf{s}) - V^*(\phi(\mathbf{s}))| \leq g(\mathbf{s}, \tilde{d}) + \frac{\gamma}{1 - \gamma} \max_{u \in \mathcal{S}} g(u, \tilde{d})$$
where $g$ is the average distance between a state and all other states in its equivalence class under the bisimulation metric $\tilde{d}$. By specifying a $\epsilon$-neighborhood for each cluster of states we can replace $g$:
$$(1 - c)|V^*(\mathbf{s}) - V^*(\phi(\mathbf{s}))| \leq 2\epsilon + \frac{\gamma}{1 - \gamma} 2\epsilon$$
$$|V^*(\mathbf{s}) - V^*(\phi(\mathbf{s}))| \leq \frac{1}{1 - c}\left(2\epsilon + \frac{\gamma}{1 - \gamma} 2\epsilon\right)$$
$$= \frac{2\epsilon}{(1 - \gamma)(1 - c)}.$$
$\square$

**Theorem 4.** *Given an encoder $\phi : \mathcal{S} \mapsto \mathcal{Z}$ that maps observations to a latent bisimulation metric representation where $||\phi(\mathbf{s}_i) - \phi(\mathbf{s}_j)||_1 := \tilde{d}(\mathbf{s}_i, \mathbf{s}_j)$, $\mathcal{Z}$ encodes information about all the causal ancestors of the reward $AN(R)$.*

*Proof.* We assume a MDP with a state space $\mathcal{S} := \{\mathcal{S}^1, ..., \mathcal{S}^K\}$ that can be factorized into $K$ variables with 1-step causal transition dynamics described by a causal graph $\mathcal{G}$ (example in Figure 10). We break the proof up into two parts: 1) show that if a factor $\mathcal{S}^i \notin AN(R)$ changes, the bisimulation distance between the original state $\mathbf{s}$ and the new state $\mathbf{s}'$ is 0. and 2) show that if a factor $\mathcal{S}^j \in AN(R)$ changes, the bisimulation distance can be $> 0$.

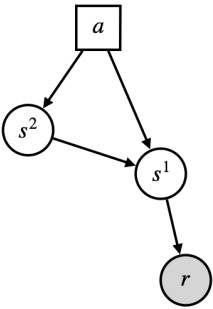

Figure 10: Causal graph of transition dynamics. Reward depends only on $\mathbf{s}^1$ as a causal parent, but $\mathbf{s}^1$ causally depends on $\mathbf{s}^2$, so AN(R) is the set $\{\mathbf{s}^1, \mathbf{s}^2\}$.

1) If $\mathcal{S}^i \notin AN(R)$, an intervention on that factor does not affect current or future reward.
$$\tilde{d}(\mathbf{s}_i, \mathbf{s}_j) = \max_{a \in A}(1 - c)|r_{\mathbf{s}_i}^{\mathbf{a}} - r_{\mathbf{s}_j}^{\mathbf{a}}| + cW(\tilde{d})(\mathcal{P}_{\mathbf{s}_i}^{\mathbf{a}}, \mathcal{P}_{\mathbf{s}_j}^{\mathbf{a}})$$

$$= \max_{a \in A} cW(\tilde{d})(\mathcal{P}_{\mathbf{s}_i}^{\mathbf{a}}, \mathcal{P}_{\mathbf{s}_j}^{\mathbf{a}}) \quad \mathbf{s}_i \text{ and } \mathbf{s}_j \text{ have the same reward.}$$

If $\mathcal{S}^i$ does not affect future reward, then states $\mathbf{s}_i$ and $\mathbf{s}_j$ will have the same future reward conditioned on all future actions. This gives us
$$\tilde{d}(\mathbf{s}, \mathbf{s}') = 0.$$
2) If there is an intervention on $S^j \in AN(R)$ then current and/or future reward can change. If current reward changes, then we already have $\max_{\mathbf{a} \in \mathcal{A}}(1 - c)|r_{\mathbf{s}_i}^{\mathbf{a}} - r_{\mathbf{s}_j}^{\mathbf{a}}| > 0$, giving us $\tilde{d}(\mathbf{s}_i, \mathbf{s}_j) > 0$. If only future reward changes, then those future states will have nonzero bisimilarity, and $\max_{\mathbf{a} \in \mathcal{A}} W(\tilde{d})(P_{\mathbf{s}_i}^{\mathbf{a}}, P_{\mathbf{s}_j}^{\mathbf{a}}) > 0$, giving us $\tilde{d}(\mathbf{s}_i, \mathbf{s}_j) > 0$. □

## B  Definition of State

Since we are concerned primarily with learning from image observations, we could explicitly distinguish the image observation space $\mathcal{O}$ from an unknown state space $\mathcal{S}$. However, since we are not tackling the general POMDP problem, we consider the *Block MDP* (Du et al., 2019), which assumes the state space is latent, and that we are instead given access to an observation space $\mathcal{O}$ and rendering function $q : \mathcal{S} \mapsto \mathcal{O}$. The crucial assumption that distinguishes the Block MDP from partially observable MDPs is the following:

**Assumption 1** (Block structure (Du et al., 2019)). *Each observation $\mathbf{o}$ uniquely determines its generating state $\mathbf{s}$. That is, the observation space $\mathcal{O}$ can be partitioned into disjoint blocks $\mathcal{O}_s$, each containing the support of the conditional distribution $q(\mathbf{o}|\mathbf{s})$.*

This assumption gives us the Markov property in the observation space $\mathbf{o} \in \mathcal{O}$. As an example, one can think of the proprioceptive state consisting of positions and velocities of actuators as the underlying state, and stacked pixel observations from a specific camera angle as a particular rendering function and corresponding observation space.

## C  Additional DMC Results

In Figure 11 we show performance on the default setting on 9 different environments from DMC. Figures 12 and 13 give performance on the simple distractors and natural video settings for all 9 environments.

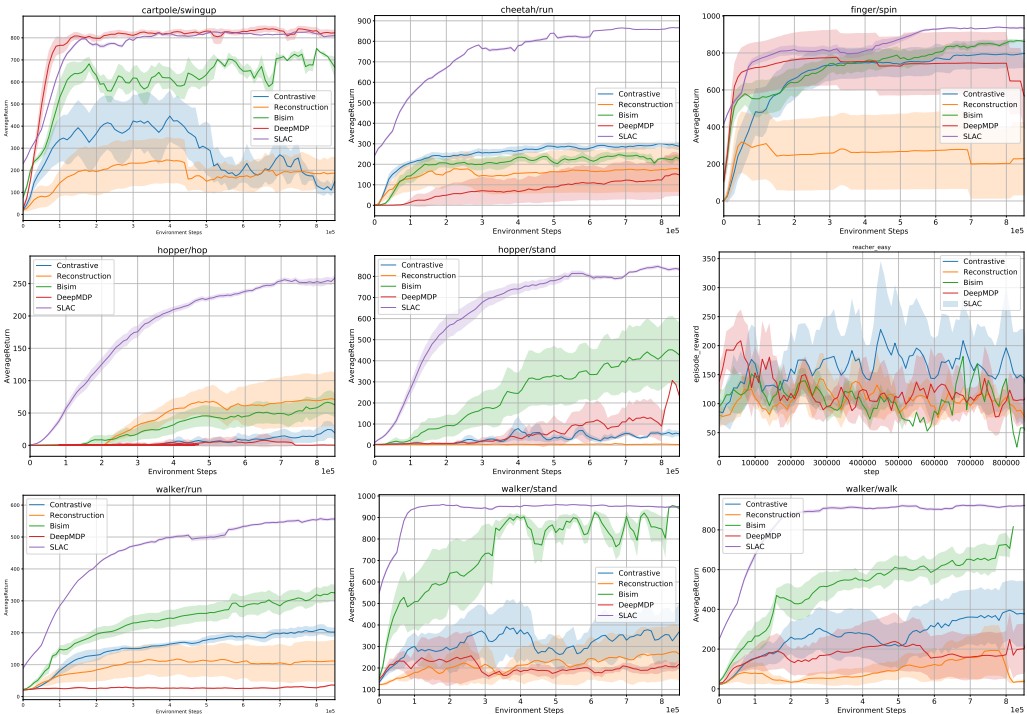

Figure 11: Results for DBC in the default setting, in comparison to baselines with reconstruction loss, contrastive loss, and SLAC on 10 seeds with 1 standard error shaded.

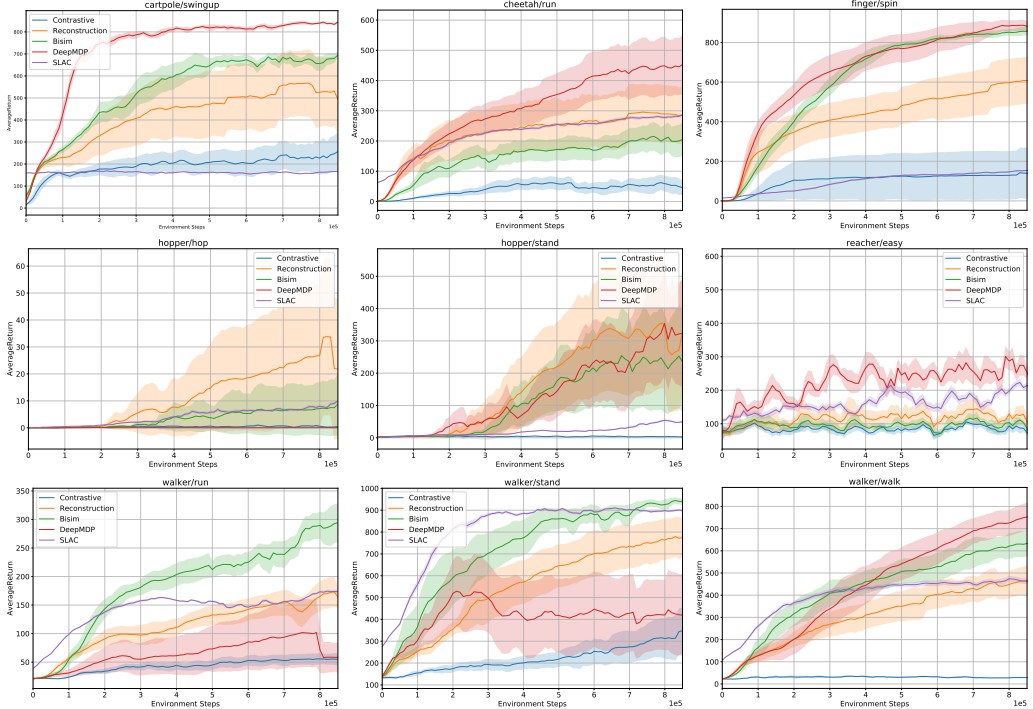

Figure 12: Results for DBC in the simple distractors setting, in comparison to baselines with reconstruction loss, contrastive loss, DeepMDP, and SLAC on 10 seeds with 1 standard error shaded.

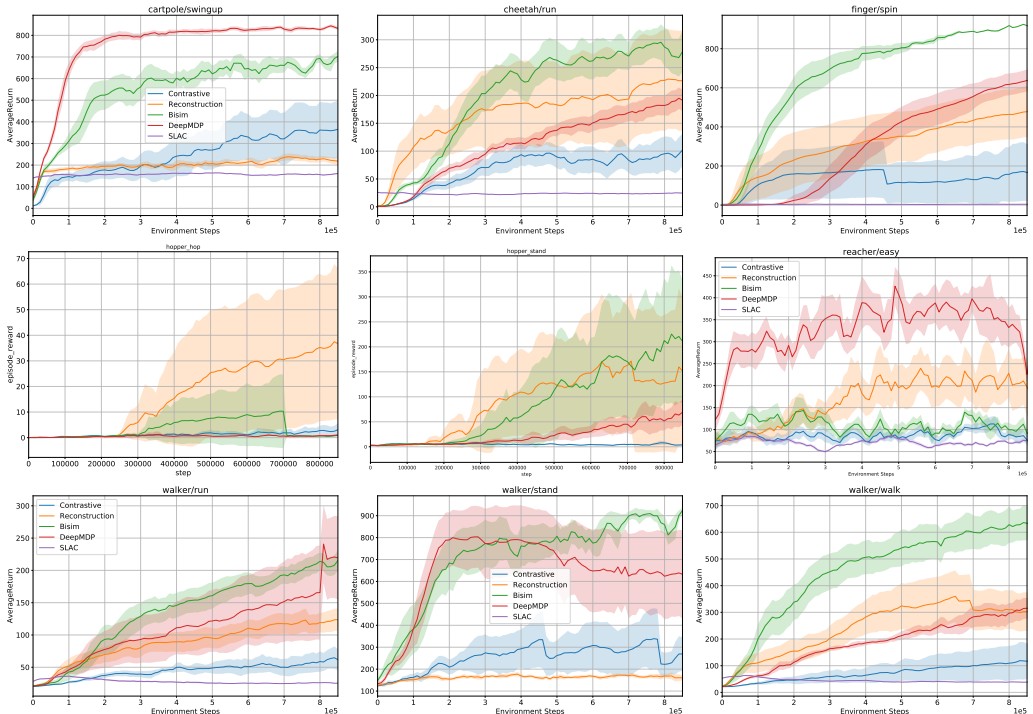

Figure 13: Results for our bisimulation metric method in the natural video setting, in comparison to baselines with reconstruction loss, contrastive loss, DeepMDP, and SLAC on 10 seeds with 1 standard error shaded.

## D   Additional Visualizations

In addition to Figure 4, we also took 10 nearby points in the t-SNE plot and average the observations, shown on the far left of Figure 14. Note the robot agent is quite crisp, which means neighboring points encode the agent in similar positions, but the backgrounds are very different, and so are blurry when averaged.

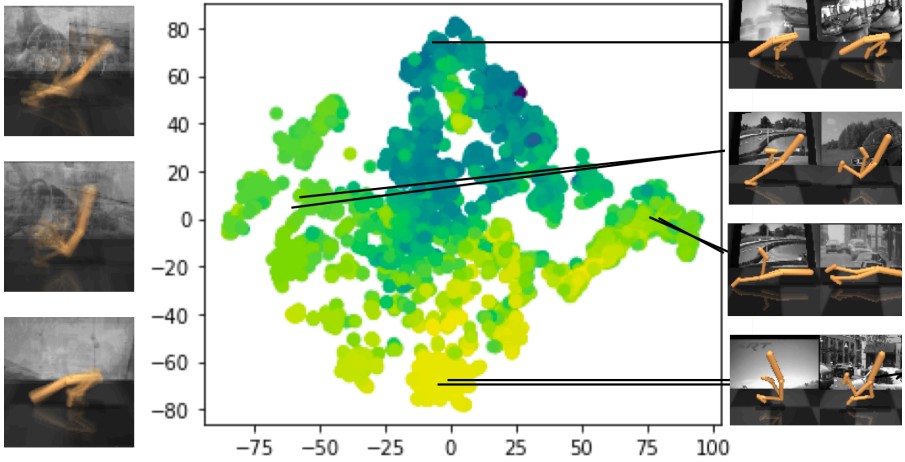

Figure 14: t-SNE of latent spaces learned with a bisimulation metric after training has completed, color-coded with predicted state values (higher value yellow, lower value purple). Neighboring points (right) in the embedding space learned with a bisimulation metric have similar encodings (middle). When we sample from the same latent point, and average the images, we see the robot configuration is crisp, meaning neighboring points encode the agent in similar positions, but the backgrounds are very different, and so are blurry when averaged.

# E  Implementation Details

We use the same encoder architecture as in Yarats et al. (2019), which is an almost identical encoder architecture as in Tassa et al. (2018), with two more convolutional layers to the convnet trunk. The encoder has kernels of size $3 \times 3$ with 32 channels for all the convolutional layers and set stride to 1 everywhere, except of the first convolutional layer, which has stride 2, and interpolate with `ReLU` activations. Finally, we add `tanh` nonlinearity to the 50 dimensional output of the fully-connected layer.

For the reconstruction method, the decoder consists of a fully-connected layer followed by four deconvolutional layers. We use `ReLU` activations after each layer, except the final deconvolutional layer that produces pixels representation. Each deconvolutional layer has kernels of size $3 \times 3$ with 32 channels and stride 1, except of the last layer, where stride is 2.

The dynamics and reward models are both MLPs with two hidden layers with 200 neurons each and `ReLU` activations.

Soft Actor Critic (SAC) (Haarnoja et al., 2018) is an off-policy actor-critic method that uses the maximum entropy framework for soft policy iteration. At each iteration, SAC performs soft policy evaluation and improvement steps. The policy evaluation step fits a parametric soft Q-function $Q(\mathbf{s}_t, \mathbf{a}_t)$ using transitions sampled from the replay buffer $\mathcal{D}$ by minimizing the soft Bellman residual,

$$J(Q) = \mathbb{E}_{(\mathbf{s}_t, \mathbf{s}_t, r_t, \mathbf{s}_{t+1}) \sim \mathcal{D}} \left[ \left( Q(\mathbf{s}_t, \mathbf{a}_t) - r_t - \gamma \bar{V}(\mathbf{s}_{t+1}) \right)^2 \right]. \tag{13}$$

The target value function $\bar{V}$ is approximated via a Monte-Carlo estimate of the following expectation,

$$\bar{V}(\mathbf{s}_{t+1}) = \mathbb{E}_{\mathbf{a}_{t+1} \sim \pi} \left[ \bar{Q}(\mathbf{s}_{t+1}, \mathbf{a}_{t+1}) - \alpha \log \pi(\mathbf{a}_{t+1} | \mathbf{s}_{t+1}) \right], \tag{14}$$

where $\bar{Q}$ is the target soft Q-function parameterized by a weight vector obtained from an exponentially moving average of the Q-function weights to stabilize training. The policy improvement step then attempts to project a parametric policy $\pi(\mathbf{a}_t | \mathbf{s}_t)$ by minimizing KL divergence between the policy and a Boltzmann distribution induced by the Q-function, producing the following objective,

$$J(\pi) = \mathbb{E}_{\mathbf{s}_t \sim \mathcal{D}} \left[ \mathbb{E}_{\mathbf{a}_t \sim \pi} [\alpha \log(\pi(\mathbf{a}_t | \mathbf{s}_t)) - Q(\mathbf{s}_t, \mathbf{a}_t)] \right]. \tag{15}$$

We modify the Soft Actor-Critic PyTorch implementation by Yarats & Kostrikov (2020) and augment with a shared encoder between the actor and critic, the general model $f_s$ and task-specific models $f_\eta^e$. The forward models are multi-layer perceptions with ReLU non-linearities and two hidden layers of 200 neurons each. The encoder is a linear layer that maps to a 50-dim hidden representation. The hyperparameters used for the RL experiments are in Table 2.

| Parameter name | Value |
|---|---|
| Replay buffer capacity | $10^6$ |
| Batch size | 128 |
| Discount $\gamma$ | 0.99 |
| Optimizer | Adam |
| Critic learning rate | $10^{-5}$ |
| Critic target update frequency | 2 |
| Critic Q-function soft-update rate $\tau_Q$ | 0.005 |
| Critic encoder soft-update rate $\tau_\phi$ | 0.005 |
| Actor learning rate | $10^{-5}$ |
| Actor update frequency | 2 |
| Actor log stddev bounds | $[-5, 2]$ |
| Encoder learning rate | $10^{-5}$ |
| Decoder learning rate | $10^{-5}$ |
| Decoder weight decay | $10^{-7}$ |
| Temperature learning rate | $10^{-4}$ |
| Temperature Adam's $\beta_1$ | 0.9 |
| Init temperature | 0.1 |

Table 2: A complete overview of used hyper parameters.

