# OpenReview forum: "Learning Invariant Representations for Reinforcement Learning without Reconstruction"
_ICLR.cc/2021/Conference — ICLR 2021 Oral_

### Official Review · AnonReviewer2 · 2020-10-29
**Deep twin network loss implementation of bisimulation-based state abstraction is robust to extraneous details in realistic visual environments**

**Rating:** 9
**Confidence:** 5

**Review:**

# Paper Summary

The paper presents a new method for embedding visual images into a state space suitable for effective control by an actor-critic style RL algorithm. They show how a previously explored idea of using a bisimulation between state abstractions and reward sequences to group states that are similar from a decision theoretic perspective can be extended to a continuous deep embedded representation using twin network style learning through standard gradient descent optimization. They call the approach Deep Bisimulation for Control (DBC).  The paper also argues for the correctness of their approach using a contraction proof and a theoretical argument for generalization to new problem domains. The approach contrasts directly with algorithms that use an autoencoder to find a compact representation by reconstructing input frames or predicting future input frames. These approaches necessarily represent enough information to reconstruct both task relevant and incidental details. Experimentally, the paper shows that performance of reconstruction-based approaches degrades by a large and significant amount when extraneous background detail is present in image frames, while the proposed method is immune. The evaluation is done on widely respected benchmark of Deep Mind MuJoCo based articulated figure simulations and a CARLA realistic image simulation of car driving which shows that focusing on task specific detail is important on realistic tasks.

# Pros and Cons

The work addresses a key problem in reinforcement learning, which is learning effective policies from unlabeled visual images.  The ability to find compact, task relevant representations of high-dimensional inputs is central to the ICLR community.

The paper covers relevant background in state abstraction in RL and it is clear how it relates to the paper's contribution.

The paper clearly explains the background of bisimulation and how the idea of grouping states according to their probable reward sequences can be practically realized by grouping states by their immediate reward and future state distributions. It is also clear in principle that a twin network can be used to induce a latent space with these properties.

The paper shows that the policy will converge and that the error will be bounded and that the representation will generalize to any problem domain in which causal dependencies are a subset of the problem domain the representation was trained on. Which is nice.

There is a nice evaluation on both MuJoCo articulated figure problems from the Deep Mind Control suite as well as the CARLA simulated driving application that uses large realistically rendered images. Comparison against state-of-the-art RL algorthims makes results convincing. I think the CARLA example is nice as it shows that practical problems without artificial augmentations have the property that the input details can be distracting if fully reconstructed. The superiority of DBC in this case really makes this point well.

The paper does not report computational load associated with their approach. Presumably due to the closed form Wasserstein approximations, we are probably looking at only a fractional increase in time for the encoder network above the policy, dynamics and reward models. So roughly 30% more??

Key figures for central results are too small to get meaningful interpretations unless enlarged by a factor of 4 or 5. Somehow this seems to go against the spirit of page limits to me.

Figure 6 and the description don't seem to be aligned, but I can imagine that it can be readily fixed.


# Recommendation

I recommend acceptance. The paper shows how to extend bisimulation principle for grouping states to continuous deep actor critic methods and provides convincing evidence on standard benchmarks that it is effective in extracting a task-relevant abstraction that is robust to noise in observations and focuses on task specific detail.



# Questions

Figure 6: it seems that the “simple_distractors” environment is different than “setting 2” natural video setting. This caption could use some reworking to get parallelism clear. The ‘ideal gas’ is the same as simple_distractors?  Also the graphs seems to be about different experimental types. Does the first experiment also use frozen encoder?  The graphs do not seem to relate to the text which talks about walker_stand, walker_run reward functions. I am confused.

Can the paper have any insight into the relative performance of their algorithm versus benchmarks algorithms across different tasks (finger spin, cheetah, walker) given that they are all stick figures?

# Feedback

I was able to work through Definition 1 and assure myself it made sense. In the end I made a small picture that helped.

There seems to be some ambiguity in notation between observations, underlying state and latent variable spaces.  For instance, in section 3, script S is defined as a state space. In section 4, the function d is defined on script S x script S which is described as an observation space.  Admittedly, the work seems to be situated in an approximately fully observable world in which states and observations are somewhat equivalent. I suspect that is why the 5-camera 300-degree suround view was necessary in the CARLA experiments.

Interesting that the paper employ stop gradients on the latent representation terms when they appear in the reward and Wasserstein terms in the loss function. This is to enforce the separateness of the optmizations in algorithm 1?

Definition 2, the bisimulation metric contains a max … so this is worst case discrepancy between the futures between state space actions and empirical actions. Was average discrepancy considered? This might be relevant later in discussion about intractability in previous approaches (section 4 paragraph 3).

Equation 4 specifies an L1 metric for the distance in abstraction space ||z_i – Z_j ||. Is there a motivation for this choice? Again trying to bound the worst error?

Theorem 4 references Theorem 5 which is not in the main text.

In section 5, ideally epsilon would be briefly described before it appears in a bound.

Figure 3 is very small... the labels – particularly the subscripts are unreadable. It may not be making a point important enough to include it. The causal variables section could be shorted in general. It is a good point but not completely surprising.

Figure 4 is a key figure to support the paper's hypothesis that deep bisimulation is effective for state abstraction in noisy images. This figure really needs to be bigger. In particular, I had to strain to read the legends to understand if the axes were different between the uncluttered video of articulated figures and the cluttered video of articulated figures with a background movie in them.  It was also hard to make out which line was which.  In particular, the “cheetah” column does not show DBC improving on cluttered video scenario --- it tops out at around 250 in both top and bottom graphs, but the scales are very different. It confuses the message a bit.

There is a statement “a single loss function would be less stable and require balancing the components”. Is this speculation or based on experience? Separate optimizations implicitly define a balance between these terms wouldn’t they? Namely equal balance?

To some degree, figure 9 is making the same argument as figure 5. Could leave this out if you were tight for space.

Figure 5: What is the first column of figures to the left? It doesn’t seem to be relevant? Otherwise, I think the figure is very effective in conveying the structured embedding space does a better job of grouping similar states. It is also very small. Drawing a white border between the figure pairs would underscore visually that there are two states in each “image”

Figure 9: I could not make sense of the images on the sides of the figure. Particularly, the images on the left side.  They seemed to be abstract geometric shapes and I could not get an intuition about what the driving scenario was.

Section 6.4 “reward highway progression an penalizes collisions”  an => and

Future work – another likely avenue for future work would be to introduce some sort of memory to handle partially observable worlds. For instance, can the agent drive a car with only a forward view if given memory? Does this break down if it does not have memory? This could either be an explicit neural memory or implicit memory such as an LSTM …  Estimating uncertainty could also be important to produce agents that can work in the real world and assess when they know what they are doing and when they do not. Another future work area is in modeling of transition distributions as something more complex than Gaussians … what if there are distinct possible futures that are equally valid: it is ok for the robot to turn left or right as long as it avoids the object straight ahead?

---

> ### Author Response · Authors · 2020-11-21
> **Thank you for your review**
>
> Thank you for the extensive constructive feedback! We have made changes to the paper in line with your suggestions. We also individually address questions and concerns below.
>
> “Key figures for central results are too small”
> We’ve used the now-additional 9th page to enlarge the images.
>
> “Figure 6: it seems that the “simple_distractors” environment is different than “setting 2” natural video setting”... “The ‘ideal gas’ is the same as simple_distractors?”
> Yes, sorry for the confusion. The “ideal gas” is the same as “simple_distractors”, but distinct from the “natural video setting”. The “ideal gas” / “simple_distractors” were only visualized in the appendix, so to clarify, we have now added them into Figure 4 (middle row), and made the terminology between “ideal gas” / “simple_distractors” consistent.
>
> “There seems to be some ambiguity in notation between observations, underlying state...”
> We agree this was ambiguous, and have updated the paper to rename “observations” as “states”, since we used frame-stacked images to approximate full observability and a Markov world state.
>
> “Interesting that the paper employ stop gradients on the latent representation terms when they appear in the reward and Wasserstein terms in the loss function. This is to enforce the separateness of the optimizations in algorithm 1?”
> Yes. We found the separation stabilized training. And the stop gradients prevented gradients flowing through the regression’s target, which otherwise resulted in NaN gradients when training the encoder. The dynamics model also needs stop gradients on it’s target to avoid adapting the encoder towards trivial solutions (e.g. always outputting a constant, for sake of accurate dynamics predictions).
>
> “Definition 2, the bisimulation metric contains a max … so this is worst case discrepancy between the futures between state space actions and empirical actions. Was average discrepancy considered?”
> Yes, first introduced by Castro 2020: as an expectation w.r.t. the online policy, which is what we also use in equation 4.
>
> “Equation 4 specifies an L1 metric for the distance in abstraction space ||z_i – Z_j ||. Is there a motivation for this choice? Again trying to bound the worst error?”
> We investigate L1 vs L2, but found no difference. We opted for L1 where large errors do not dominate as much as in L2. We are not trying to bound the worst case error, only the averaged w.r.t. the online policy in equation 4, for reason of being able to use the actions stored in the replay buffer.
>
> “Theorem 4 references Theorem 5 which is not in the main text.”
> Added.
>
> “In section 5, ideally epsilon would be briefly described before it appears in a bound.”
> We have added additional discussion on epsilon, copied here for convenience. It is the amount of “allowed” approximation, where large epsilon leads to a more compact bisimulation partition at the expense of a looser bound on the optimal value function.
>
> “There is a statement “a single loss function would be less stable and require balancing the components”. Is this speculation or based on experience? Separate optimizations implicitly define a balance between these terms wouldn’t they? Namely equal balance?”
> Possibly yes, but on reflection we’ve now removed the point about “balance”, since the point about training stability was much more critical.
>
> “Figure 5: What is the first column of figures to the left? It doesn’t seem to be relevant?”
> Agreed, removed.
>
> “Drawing a white border between the figure pairs would underscore visually that there are two states in each “image””
> Agreed, done.
>
> “Figure 9: I could not make sense of the images on the sides of the figure. Particularly, the images on the left side. They seemed to be abstract geometric shapes and I could not get an intuition about what the driving scenario was.”
> These shapes were a stone wall the car had hit (during training) causing it to flip onto it’s left side, causing the sky to appear on the right side of the image. We’ve updated the caption text and main text about the CARLA task.
>
> “Future work – another likely avenue for future work would be to introduce some sort of memory to handle partially observable worlds.”
> Thanks for the additional examples! We have included them on page 9.

---

### Official Review · AnonReviewer3 · 2020-10-31
**Promising approach to learn task relevant representations for control**

**Rating:** 7
**Confidence:** 3

**Review:**

**Summary**
The paper focuses on how learning state-representations that encode information relevant to the task can improve reinforcement learning from pixels. Often, observations in an MDP can contain information that are irrelevant (“distractors”) to the task at hand and can likely “distract” the downstream RL algorithm used. Unlike existing reconstruction based approaches (which don’t explicitly incentivize ignoring task-irrelevant information), the authors propose Deep Bisimulation Control (DBC) that relies on bi-simulation metrics (as the task-aware criterion) that encode behavioral similarity b/w states with respect to the reward structure. Instead of explicitly learning a bi-similarity distance function, authors enforce the representations which under L1 distances correspond to bi-simulation metrics. DBC demonstrates learning these representations in conjunction with the control policy, reward and a dynamics model. Furthermore, the authors highlight connections to causal inference which can hopefully further provide insights into which “new” reward structures can the learned representations generalize to (since bi-similarity metrics themselves are heavily dependent on the reward structure). Results obtained by the authors demonstrate that DBC can learn task specific representations and a control policy in a robust manner in the presence of distractors on the Deepmind Control Suite and the CARLA simulator. Additionally, the authors also demonstrate how DBC can generalize to new reward functions on Mujoco.

**Strengths**

- The paper is generally well-written and easy to follow for the most part. The authors do a good job of walking the reader through the preliminaries, highlighting distinctions with prior usage of bisimilarity metrics and how DBC ties in with slight modifications to the Soft Actor-Critic algorithm.
- The results obtained on DM control suite demonstrate that while DBC is competitive or slightly worse compared to other approaches in the absence of distractors, it performs better compared the set of baselines when distractors are introduced — significantly outperforming the reconstruction and the contrastive approaches. Furthermore, compared to a VAE, DBC places observations that are similar in terms of task information closer in the learnt embedding space.
- The authors also show that compared to an adaptation of prior usage of bisimulation metrics, DBC is more sample efficient. Additionally, within the considered family of reward functions to generalize to “walker_run” and “walker_stand”, DBC shows stronger generalization compared to DeepMDP. Furthermore, learned embeddings on CARLA demonstrate qualitatively that DBC learns to group behaviorally similar distractor states (manifesting as obstacles).

**Weaknesses**
I don’t have any major weaknesses to point out. I will highlight minor comments / weaknesses which, I think if addressed, would definitely make the paper stronger.
- While the authors state that DBC in practice can be combined with any model-free of model-based algorithm, the paper would definitely benefit if this claim was backed up with results demonstrating DBC combined with another model-free or model-based algorithm since it might be unclear off-the-shelf if an algorithm requires major or minor changes to work in conjunction with DBC.
- Task generalization (or generalizing over different reward functions) relies on the assumption that the new reward function depends on contributing factors that likely influence the earlier “seen” reward function as well. While empirically validating this claim from walker_walk -> walker_run / walker_stand is a reasonable starting point, demonstrating another instance where inferring the source and target reward functions is not immediately obvious would definitely benefit the paper.

---

> ### Author Response · Authors · 2020-11-21
> **Thank you for your review**
>
> We thank the reviewer for their assessment and constructive suggestions. We address the two suggestions below -- we are not sure what the reviewer had in mind as an experiment for the second suggestion, and are happy to discuss possible tasks.
>
>
> 1. "the paper would definitely benefit if this claim was backed up with results demonstrating DBC combined with another model-free or model-based algorithm since it might be unclear off-the-shelf if an algorithm requires major or minor changes to work in conjunction with DBC."
> RL is brittle and difficult to get working -- so this takes time. We will note that as a follow up to this work we are combining DBC with Dreamer [1] and using the latent model for planning. We have initial results that show this works better than Dreamer with the original reconstruction loss. This result will be part of a new paper that is not yet finished, so we will not provide those results as part of this publication.
>
>
>
> 2. "While empirically validating this claim from walker_walk -> walker_run / walker_stand is a reasonable starting point, demonstrating another instance where inferring the source and target reward functions is not immediately obvious would definitely benefit the paper."
> As the reviewer notes, we have a theoretical result (Theorem 4) that dictates what family of reward functions our learned DBC representation can generalize to. We are not sure what "not immediately obvious" additional transfer setting the reviewer has in mind, perhaps an example would help?  We know what transfer settings would not work -- for example, if in the original task a robot receives a reward for picking up blue blocks, it will not generalize to red blocks because any red blocks  in the environment will have been dropped by the representation as irrelevant to the original task.
>
>
>
>
>
> 1. Dream to Control: Learning Behaviors by Latent Imagination. Danijar Hafner, Timothy Lillicrap, Jimmy Ba, Mohammad Norouzi. ICLR 2019.

---

### Official Review · AnonReviewer1 · 2020-10-31
**The paper is well written and presents a clear approach to a well motivated problem with strong evaluation results.**

**Rating:** 7
**Confidence:** 4

**Review:**

The authors propose an approach to robust representation learning of observations for reinforcement learning by training a model to align the euclidean distance between two observations with bisimulation metrics that quantify how similar the states that generated the observations are in terms of the control problem.  This reduces the effect of irrelevant features in the observations on the representations.

The paper is well written, the problem is clearly motivated and the approach and technical contribution is easy to follow.

The approach to use the state bisimulation metric to supervise observation representation is intuitive and clearly motivated.  Theoretical analysis is provided with generalization guarantees.

"As an example, in the context of autonomous driving, an intervention can be a change in weather, or a
change from day to night which affects the observation space but not the dynamics or reward."  I do not agree with the example as weather can directly alter the dynamics and desired behavior of an AV system.  The point in this paragraph is still clear but I would suggest a different example.

The evaluations are strong and run on a number of different experiment settings against multiple strong SOTA models.

In Figure 4 the proposed approach is outperformed by contrastive learning in the default setting.  I understand that the goal is to learn robust representations for the natural setting, but can the authors comment on why it fails to beat the contrastive approach here and provide insight on how this may be addressed.  Some problems have only a few distractors and may fall between the natural and the default setting.

recommendation and reasoning

The paper is well written and presents a clear approach to a well motivated problem with strong evaluation results.  I recommend acceptance.

---

> ### Author Response · Authors · 2020-11-21
> **Thank you for your review**
>
> Thank you for your positive review. We have addressed your two main concerns below and in the updated version of the paper.
>
> “I do not agree with the example as weather can directly alter the dynamics and desired behavior of an AV system”
> Yes, we agree (e.g. wet roads can cause sliding), so we updated our example to day/night lighting which affects the observation space but not the physical dynamics”.
>
> “In Figure 4 the proposed approach is outperformed by contrastive learning in the default setting. I understand that the goal is to learn robust representations for the natural setting, but can the authors comment on why it fails to beat the contrastive approach here and provide insight on how this may be addressed.”
> In the default setting there are no equivalent observations to be compressed. Therefore the contrastive approach performs correctly (whereas it does not when there are bisimilar observations -- two bisimilar observations can end up as the negative pair with a contrastive loss, so it does not compress anything). Therefore, this result shows that when nothing needs to be compressed, a contrastive loss is easier to optimize than matching exact distances in a latent space (a regression problem).

---

### Decision · Program_Chairs · 2021-01-07
**Final Decision**

**Decision:**

Accept (Oral)

**Comment:**

This paper proposed using the state bisimulation metric to learn invariant representations for reinforcement learning.  The method is generic, effective, and is supported by both theoretical and experimental results.  All reviewers and I think this is a strong contribution to the area.